# Observed trends and projections of temperature and precipitation in the Olifants River Catchment in South Africa

**Abiodun Morakinyo Adeola**[1,2,3]*, **Andries Kruger**[1,3], **Thabo Elias Makgoale**[1],
**Joel Ondego Botai**[1,3,4]

**1** South African Weather Service, Pretoria, South Africa, **2** School for Health Systems and Public Health, University of Pretoria, Pretoria, South Africa, **3** Department of Geography, Geoinformatics and Meteorology, University of Pretoria, Pretoria, South Africa, **4** School of Agricultural, Earth and Environmental Sciences, University of KwaZulu-Natal, Westville Campus, Durban, South Africa

* abiodun.adeola@weathersa.co.za

**Data Availability Statement:** The data process is done using R Software which includes several packages for mapping NetCDF data. Sample data and scripts used for this study are made available

## Abstract

Among the projected effects of climate change, water resources are at the center of the matrix. Certainly, the southern African climate is changing, consequently, localized studies are needed to determine the magnitude of anticipated changes for effective adaptation. Utilizing historical observation data over the Olifants River Catchment, we examined trends in temperature and rainfall for the period 1976–2019. In addition, future climate change projections under the RCP 4.5 and RCP 8.5 scenarios for two time periods of 2036–2065 (near future) and 2066–2095 (far future) were analysed using an ensemble of eight regional climate model (RCA4) simulations of the CORDEX Africa initiative. A modified Mann-Kendall test was used to determine trends and the statistical significance of annual and seasonal rainfall and temperature. The characteristics of extreme dry conditions were assessed by computing the Standardized Precipitation Index (SPI). The results suggest that the catchment has witnessed an increase in temperatures and an overall decline in rainfall, although no significant changes have been detected in the distribution of rainfall over time. Furthermore, the surface temperature is expected to rise significantly, continuing a trend already evident in historical developments. The results further indicate that the minimum temperatures over the Catchment are getting warmer than the maximum temperatures. Seasonally, the minimum temperature warms more frequently in the summer season from December to February (DJF) and the spring season from September to November (SON) than in the winter season from June to August (JJA) and in the autumn season from March to May (MAM). The results of the SPI affirm the persistent drought conditions over the Catchment. In the context of the current global warming, this study provides an insight into the changing characteristics of temperatures and rainfall in a local context. The information in this study can provide policymakers with useful information to help them make informed decisions regarding the Olifants River Catchment and its resources.

on Open Science Framework at https://osf.io/8rhn2
and https://osf.io/3d9j4.

**Funding:** Partial funding for the background
research was received from the South32 mining
company. There was no additional external funding
received for this study. The funders had no role in
study design, data collection and analysis, decision
to publish, or preparation of the manuscript.

**Competing interests:** The authors have declared
that no competing interests exist.

## Introduction

Sub-Saharan Africa (SSA) is more likely to be affected by climate change than any other region
because of its high exposure and limited capability for adaptation [1]. Climate change is
expected to exacerbate current food insecurity, health challenges, poverty, and development
challenges in SSA [2]. SSA is one of the regions where adaptation planning has been hindered
by the complexity and relative uncertainty surrounding climate change impacts [3]. While
uncertainties surround the understanding of future climate impacts in most regions of the
globe, the SSA has among the lowest levels of confidence due to fewer research studies con-
ducted using fine resolution climate models [1]. Modelling with a finer resolution will contrib-
ute to enhancing the level of certainty which is crucial to planning an effective adaptation
strategy [1].

Southern Africa is particularly shown to be susceptible to climate change because of its sig-
nificant economic inequality and strong dependency on rain-fed agriculture [4]. In the face of
global warming caused by emissions of greenhouse gases, it is widely anticipated that extreme
weather events will become more frequent, intense and severe [5]. South Africa, like other
countries in the southern region, has recently witnessed rises in the frequency and intensity of
extreme weather conditions such as droughts and floods. These weather systems left behind
them huge traces of economic destruction including losses of life and properties [6].

The impacts of climate change are expected to worsen current challenges in many climate-
sensitive sectors, increasing their vulnerability as well as the human systems that depend on
them [7, 8]. Consequently, studies have been done on analysing possible future projections of
climatic variables over the near- to far-term [9]. As the climate warms from current levels to
near, medium, and long-term futures, it is crucial to understand the extent to which climate
and extreme weather events will change at the local level for better preparedness and the for-
mulation of effective policy [9]. Globally, the overall impacts of climate change on freshwater
resources are projected to be negative.

Climate change, coupled with increasing population, and the demand for social and eco-
nomic equity all present challenges that need to be identified and addressed. Hence, by under-
standing the dynamics of current climate variability and future change, effective resource
management, planning, and adaptation can be achieved [10]. In order to achieve enhanced
capacity and adaptation strategies to cope with these inherent dynamics, climate change stud-
ies are needed at a local scale (fine resolution) [11–13]. This builds the capacity to adapt to
future climate change and strengthens resilience to current climate challenges [10].

Studies have found that to make the most effective decisions for policy formulation on cli-
mate change adaptation and climate services, short-term and mid-term climate projections at
a local scale are needed [14]. Climate change projections have so far been conducted at the
regional scale in Southern Africa and SSA in particular, using general circulation models
(GCMs) [15, 16]. Studies have demonstrated that GCMs are capable of reproducing tempera-
ture distributions realistically. However, they are liable to overestimating precipitation over
Southern Africa for all seasons [17, 18]. Furthermore, whilst GCMs are relatively good at pro-
ducing projections over a wide region, the determination of local impacts and the development
of effective adaptation measures requires a thorough knowledge of the local conditions [19].
Consequently, studies have begun using regional downscaling techniques as a means of achiev-
ing greater levels of detail and improving projection accuracy [20–22]. For instance, [17] dem-
onstrated that Regional Climate Models (RCMs) can improve the accuracy of climate
projections, especially in places where the terrain is highly heterogeneous with the small-scale
climatic system. In addition, [23, 24] showed that an ensemble of ten RCMs effectively simu-
lates precipitation distribution patterns over Southern and Eastern Africa respectively.

Further, [22] indicated that the downscaled Unified Model by the Met Office over the Africa domain enabled the inclusion of additional details, such as higher resolutions and convection parametrizations. In their study, they observed that the addition of convection enhanced rainfall simulation from June to August. Typical GCM-based models have inherent biases within their boundary conditions; conversely, [18] demonstrated that forcing the GCMs with local climate data reduces these biases over the African region. Additionally, studies have shown that RCMs also provide an improved representation of annual cycles in Southern Africa with a greater level of detail in projections [12, 18, 25].

This study aims at investigating the trends in historical maximum and minimum temperature and rainfall, the estimation of plausible future climate changes, as well as the magnitude of future occurrences in the variables to inform adaptation initiatives over the Olifants River Catchment. As mentioned, this analysis can provide a basis for future impact analysis and the development of effective adaptation strategies for various climate-sensitive sectors within the catchment.

## Materials and methods

### Study area

The Olifants River Catchment is located in the Limpopo River Basin, a regional drainage basin extending across South Africa, Mozambique, Zimbabwe and Botswana (Fig 1). About 3.2 million people live in the basin's 54,475 $km^2$, of whom two-thirds live in rural areas that transverse the parts of Ekurhuleni, Sedibeng and City of Tshwane districts of Gauteng province, the Mopani, Capricorn, Waterberg and Sekhukhune districts of Limpopo province and Gert Sibande, Nkangala and Ehlanzeni districts of Mpumalanga province [26]. About 40% of the water in the Limpopo River originates from the Olifants River, making it an integral part of the drainage system. The river flows for about 560 km from Gauteng, through the Mpumalanga and Limpopo Provinces of South Africa through Mozambique and into the Indian Ocean. Mining, industry, and agriculture are major economic activities within the catchment and account for about 6% of the gross domestic product of South Africa [27]. A large portion of the catchment receives an average annual rainfall of 500 to 800 mm from October to April but reaches more than1000 mm along the escarpment [28]. Elevations within the catchment range from 200m to about 2300m above sea level, and therefore year-round temperatures have a large range, between about 0˚C to 35˚C.

### Data set

The climate data was acquired from the South African Weather Service (SAWS) climate database. There are four climate stations within the proximity of the catchment which measure temperature and rainfall and have sufficiently long data records, i.e., 1976–2019. The data were quality controlled such that any station with 20% or more of the total possible values missing, was excluded from the analysis. Consequently, two stations with data for the period of 1976–2019 were deemed sufficient for analysis over the study area given the data quality. The climate data analysed include the daily rainfall, minimum temperature (Tmin) and maximum temperature (Tmax). Locations of the stations within the study area and the completeness of their records are shown in Fig 1 and Table 1.

In addition, the SAWS district rainfall dataset was analysed. The dataset is a delineation of areas with homogeneous rainfall, in which the entire nation is divided into a total of 94 rainfall districts [32], with the relevant districts indicated in Fig 1. These districts are mainly defined by the annual march of monthly maximum rainfall, the boundaries between winter and summer rainfall regions as well as the topography. A district rainfall total is determined by taking

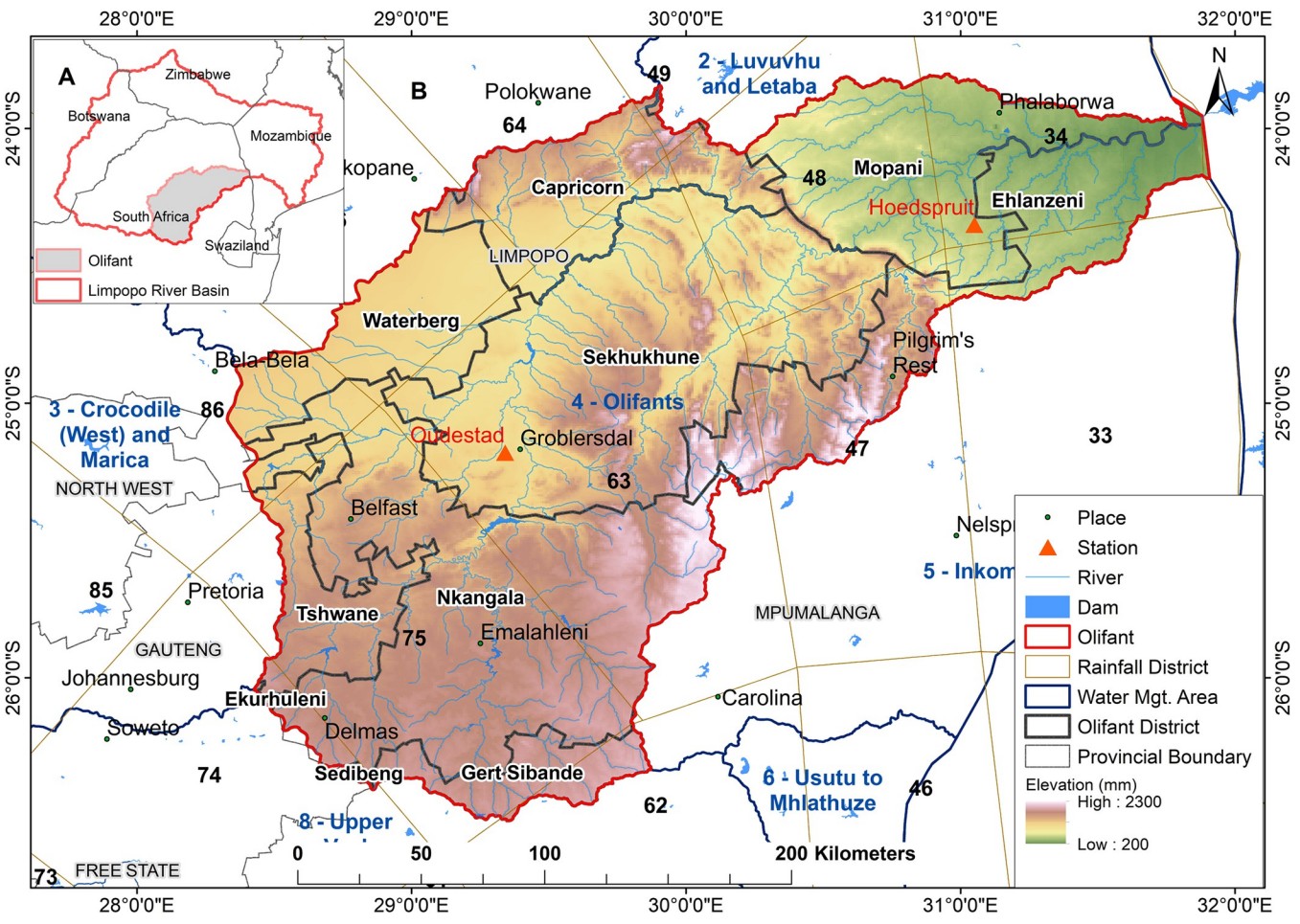

**Fig 1. Location of temperature stations and delineation of numbered homogenous rainfall regions used for analysis.** The inset map indicates the location of Olifants River Catchment in South Africa. The Place, Station and Rainfall District shapefile in the Figure were acquired database of the South African Weather Service. The River, Dam, Water management shapefile were collected from the South Africa National Department of Water and Sanitation available from [29]. The Geographic administrative shapefile was collected from the Demarcation board of South Africa available from [30]. The elevation data was downloaded from [31].

the average of the daily rainfall values of available or operational rainfall stations in the district monthly [33]. For this study, the relevant rainfall districts can be considered to be topographically less complex than most of the rainfall districts in South Africa, which makes the reliability of the monthly statistics relatively representative. Consequently, rainfall totals over two rainfall districts (34 and 63) in which the selected climate stations are located, were analysed. Data sets are available upon request at https://www.weathersa.co.za/home/equiries_climatedata.

Additionally, an ensemble of eight individual Global Climate Models (GCMs) of the Fifth Phase of Coupled Model Inter-comparison Project (CMIP5) (Table 2) was dynamically downscaled to a spatial resolution (0.44˚ x 0.44˚) by the Rossby Centre regional (RCA4) model was

**Table 1. Summary of temperature stations with location and elevation.**

| Name | Latitude (degrees decimal) | Longitude (degrees decimal) | Elevation (a.m.s.l.) (m) | Start Year of analysis |
|---|---|---|---|---|
| Hoedspruit | -24.35 | 31.05 | 520 | 1976 |
| Oudestad | -25.18 | 29.34 | 955 | 1976 |

**Table 2. Summary of the Global Circulation Models (GCMs) used in the study.**

| Model name | Country | Resolution | Literature |
|---|---|---|---|
| CanESM2m | Canada | 2.8˚ x 2.8˚ | [36] |
| CNRM-CM5 | France | 1.4˚ x 1.4˚ | [37] |
| CSIRO-Mk3 | Australia | 1.9˚ x 1.9˚ | [38] |
| IPSL-CM5A-MR | France | 1.9˚ x 3.8˚ | [39] |
| MICRO5 | Japan | 1.4˚ x 1.4˚ | [40] |
| MPI-ESM-LR | Germany | 1.9˚ x 1.9˚ | [41] |
| NorESMI-M | Norway | 1.9˚ x 2.5˚ | [42] |
| GFDL-ESM2M | USA | 2.0˚ x 2.5˚ | [43] |

used for the future projections. The RCA4 simulated projections are part of the Coordinated Regional Climate Downscaling Experiment (CORDEX) [34, 35]. For this study, the selection of the eight models is based on computational resources and data completeness.

In this study, the climate change projections were performed based on two climate change scenarios. The Representative Concentration Pathways (RCP) 4.5 indicates a 4.5 watts per metre squared–W/m$^2$ forcing increase relative to pre-industrial conditions and the RCP8.5 indicates a 8.5 watts per metre squared–W/m$^2$ forcing increase relative to pre-industrial conditions. The RCP4.5 assumes a medium stabilization scenario while the RCP8.5 assumes a scenario that includes no policy-driven mitigation (also refers to as a "business-as-usual" scenario) [44].

## Analysis

**Statistical and trend analysis.** The daily maximum, minimum temperature and rainfall data were aggregated to monthly, seasonal and annual data. A boxplot analysis was used to examine the general statistical characteristics of the data. The annual and seasonal time series were analyzed for trends using the Mann-Kendall (MK) test statistic *S* presented in Eq 1 [45]. The MK is a non-parametric test that makes no assumptions about how the underlying data will be distributed and is relatively insensitive to outliers [45].

$$S = \sum_{k=1}^{n-1} \sum_{j=k+1}^{n} sgn(x_j - x_k) \tag{1}$$

$$sgn(x_j - x_k) = \begin{cases} 1 \text{ if } x_j - x_k > 0 \\ 0 \text{ if } x_j - x_k = 0 \\ -1 \text{ if } x_j - x_k < 0 \end{cases}$$

where, $x_j$ and $x_k$ are the annual or seasonal values in years *j* and *k* respectively.

An upward or a downward trend is indicated when S is positive or negative respectively. To determine whether a trend is statistically significant, the Z value is evaluated. Z values in the positive range indicate upward trends and Z values in the negative range indicate downward trends. The 95% level of confidence was used to test whether there was an upward or downward monotone trend (a two-tailed test). Hence, for statistical significance, the probability estimate must be equal to or less than 0.05 in a 95% confidence interval.

The change per unit time (annual or seasonal) within the time series was estimated using Sen's nonparametric approach with an assumption that the trend is linear [46]. The magnitude of the trend is predicted by Sen's estimator. The slope is computed using Eq 2. A positive or

negative value of $Q_i$ indicates an increasing or decreasing rate, respectively.

$$Q_i = \frac{(x_j - x_k)}{j - k} \, for \, i = 1, 2, 3 \ldots \tag{2}$$

Where, $x_j$ and $x_k$ are the annual or seasonal values in years $j$ and $k$ respectively.

**Standardized Precipitation Index (SPI).**   For further analysis, the SPI was computed to verify the historical variations in extreme features such as drought. The SPI is s drought index commonly used to describe dry conditions over a wide range of timescales. Drought events begin when the values of SPI, at any timescale being investigated, become continuously negative and reach a value of -1 [47]. The SPI was calculated based on [47] for a 2-, 6-, 12-month accumulation period. For application purposes, 1- or 2-month SPI is applicable for meteorological drought, while consistent dry conditions from 2-month to 6-month are considered agricultural drought, and if continuous from 6-month up to 24-month or more it is categorized as hydrological drought. In particular, the SPI was calculated by fitting a gamma distribution described by Eq 3, to monthly rainfall data for districts 34 and 63 (Fig 1).

$$g(x) = \frac{1}{\beta^\alpha \Gamma(\alpha)} x^{\alpha-1} e^{-x/\beta} \tag{3}$$

where $\alpha > 0$ represents the shape parameter, $\beta > 0$ is a scalar parameter, $x > 0$ represents the amount of rainfall and $\Gamma(\alpha)$ is the gamma function given by Eq 4.

$$\Gamma(\alpha) = \int_0^\infty y^{a-1} e^{-y} dy \tag{4}$$

where $n$ is the number of observations. The gamma distribution in Eq 1 was used to compute the cumulative probability function, as per Eq 5,

$$G(x) = \int_0^x g(x) dx = \int_0^x \frac{1}{\beta^\alpha \Gamma(\alpha)} x^{\alpha-1} e^{-x/\beta} dx = \frac{1}{\Gamma(a)} \int_0^x t^{a-1} e^{-1} dt \tag{5}$$

The cumulative probability function (Eq 5) was then transformed into the standard normal distribution to yield the SPI, a time series consisting of both negative and positive values considered as dry and wet conditions, respectively. The resulting SPI values were categorized based on the classification in Table 3.

**Model evaluation.**   The reliability of the individual model and ensembles to simulate observed annual and seasonal data was assessed using the Taylor diagram [48]. The diagram is a visual representation of how closely a simulated pattern fits observation. The Taylor diagram is quantified in terms of the correlation ($R$), the centred root-mean-square-error ($RMSE$) and the amplitude of the standard deviations ($SD$). Moreover, the uncertainty of the different

**Table 3. SPI and EDI values range for drought [44].**

| Category | SPI Values |
|---|---|
| **Extremely Dry** | $\leq$-2.0 |
| Severely Dry | -1.5 to -1.99 |
| Moderately Dry | -1.0 to -1.49 |
| Normal | -0.99 to 0.99 |
| Moderately Wet | 1.0 to 1.49 |
| Severely Wet | 1.5 to 1.99 |
| Extremely Wet | $\geq$2.0 |

multi-model ensembles is also compared. The three statistics are integrated by Eq 6;

$$E' = \sigma_o^2 + \sigma_s^2 - 2\sigma_o\sigma_s\rho \tag{6}$$

where $E'$ is the centred root-mean-square-error, $\sigma_o$ is the standard deviation of observed values, $\sigma_s$ is the standard deviation of simulated values and $\rho$ is the correlation coefficient between the simulated and observed values.

For a given linear analysis of the model, a model with low variance and low bias is considered the ideal model and most suitable for projection while a model with high variance and low bias is considered an overfitting model; and a model with low variance and high bias is typically regarded as an underfitting model. In addition, a model with high variance and high bias will highly probably give the greatest prediction error.

**Climate change projections.** Daily simulated total rainfall, and maximum and minimum temperature averages are used to generate projections of annual and 3-month seasonal change. Future projections of rainfall, Tmax and Tmin are presented for the two 30-year periods of 2036–2065 (near future) and 2066–2095 (far future) under the RCP 4.5 and RCP8.5 scenarios. Projected changes are expressed relative to the historical 30-year period of 1976 to 2005. The seasons considered are December-January-February (DJF), March-April-May (MAM), June-July-August (JJA) and September-October-November (SON). Additionally, the Probability Distribution Function was used to estimate the occurrence (number of possible events) of both the historical and projected Tmax, Tmin and rainfall generated over the entire catchment.

## Results

### Historical variations

Fig 2 shows box plots of Tmax, Tmin and rainfall indicating the values for minimum, first quartile, median, third quartile, and maximum for the two weather stations and corresponding

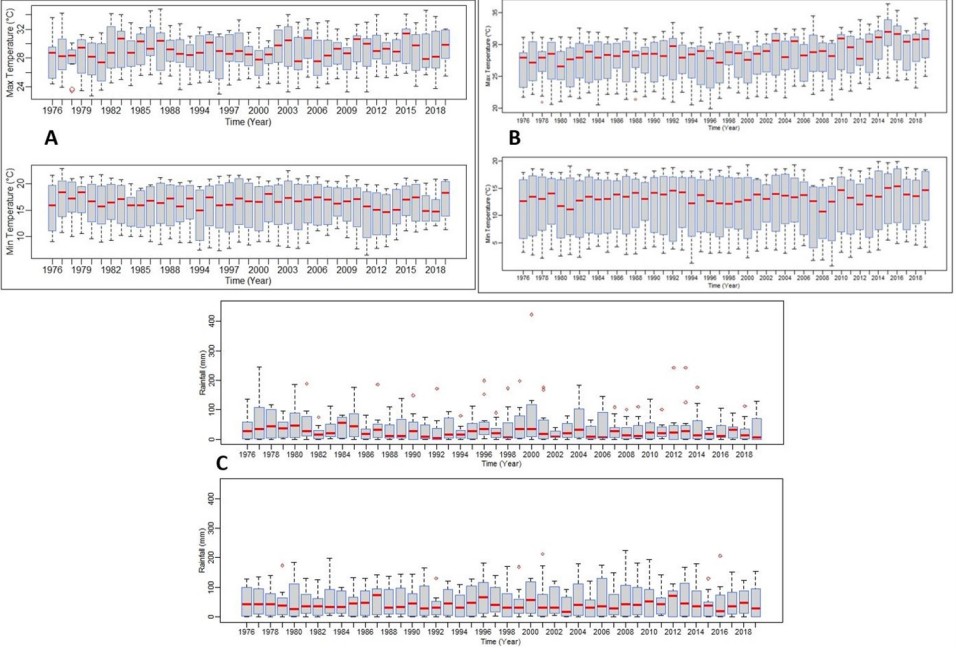

**Fig 2.** Boxplot of minimum temperatures (A top; B top) and maximum temperature (A bottom; B bottom) over Hoedspruit (A) and Oudestad (B) stations and rainfall (C) top District 34 and bottom District 63.

homogenous rainfall districts for 1976–2019. At a monthly average temperature (Tmean) of 25.9 and 24.5˚C January are the hottest month of the year over the northern part (Hoedspruit station) and southern part (Oudestad station) of the catchment respectively. At 17.3 and 13.2˚C on average, the month of July is the coldest of the year over the northern part and southern part of the catchment respectively. The southern part of the catchment is wetter with larger rainfall received in January. The difference in rainfall between the wettest and driest months is 80.0 mm and 103.6 mm over Hoedspruit and Oudestad stations respectively (Table 4).

## Trends

The descriptive statistics and historical trends of annual and seasonal rainfall, Tmax and Tmin for the two climate stations and corresponding homogenous rainfall districts with long term data are presented on seasonal and annual bases from 1976–2019 in Table 5. In the northern part of the catchment at Hoedspruit station, Tmean ranged between 27.3 to 30.2˚C with an average value of 28.8 ± 0.7˚C for the period 1976 to 2019. The Mann-Kendall trend analysis showed that test Z was +0.201 and Sen's slope estimator was +0.018, which indicated that the mean annual Tmax significantly increased at the rate of 0.018˚C per year at the 95% confidence level over the analysis period. On the other hand, the mean annual Tmin varied between 14.3 to 17.8˚C with a mean value of 15.9 ± 0.7˚C (Table 5). The trend analysis of mean annual Tmin showed that it decreased with a Z = -0.086 at the rate of -0.01˚C per year for the period of study. Therefore, unlike Tmax with abs(Z) < 0.05, there is no significant trend in Tmin. Seasonally, Tmax shows an increasing trend on an annual basis. Seasonally, however, the increasing trend is only statistically significant in winter and spring. A decreasing trend in Tmin is evident for all the seasons except for winter, however, only the summer trend of Tmin is statistically significant. Overall, Tmean has increased by 0.4˚C in the northern part. Although the rainfall shows a decreasing trend, the annual and seasonal trends are not statistically significant.

In the southern part of the catchment at the Oudestad climate station, the mean annual Tmax ranged between 26.1 to 31.3˚C with an average value of 27.9 ± 1.1˚C for the period 1976 to 2019. The Mann-Kendall trend analysis showed that test Z was +0.53 and Sen's slope estimator was +0.055, which indicated that the mean annual Tmax significantly increased at the

**Table 4. Mean monthly values of Maximum temperature (Tmax), Minimum temperature (Tmin), Average Temperature (Tmean) and total rainfall over the period of study (1976–2019).**

| Month | Hoedspruit/District 34 | | | | Oudestad/District 63 | | | |
|---|---|---|---|---|---|---|---|---|
| | Tmax (˚C) | Tmin (˚C) | TMean (˚C) | Rainfall (mm) | Tmax (˚C) | Tmin (˚C) | TMean (˚C) | Rainfall (mm) |
| Jan | 31.4 | 20.4 | 25.9 | 83.0 | 31.1 | 17.9 | 24.5 | 106.3 |
| Feb | 31.3 | 20.3 | 25.8 | 79.1 | 31.2 | 17.5 | 24.3 | 79.1 |
| Mar | 30.6 | 19.3 | 25.0 | 52.8 | 30.0 | 16.0 | 23.0 | 71.7 |
| Apr | 28.6 | 16.5 | 22.6 | 26.3 | 27.7 | 12.1 | 19.9 | 32.3 |
| May | 27.2 | 12.8 | 20.0 | 8.4 | 25.4 | 7.7 | 16.5 | 11.2 |
| Jun | 25.3 | 10.0 | 17.7 | 2.9 | 22.7 | 4.3 | 13.5 | 4.3 |
| Jul | 24.7 | 9.8 | 17.3 | 4.8 | 22.6 | 3.8 | 13.2 | 2.7 |
| Aug | 26.6 | 11.8 | 19.2 | 4.9 | 25.3 | 6.4 | 15.8 | 6.3 |
| Sep | 28.6 | 14.5 | 21.6 | 10.0 | 28.7 | 10.9 | 19.8 | 17.1 |
| Oct | 29.3 | 16.6 | 22.9 | 29.3 | 29.8 | 14.4 | 22.1 | 64.3 |
| Nov | 30.4 | 18.4 | 24.4 | 64.0 | 30.0 | 16.1 | 23.1 | 105.0 |
| Dec | 31.4 | 19.7 | 25.6 | 81.1 | 30.8 | 17.5 | 24.1 | 119.7 |

**Table 5. Descriptive statistics and trend analysis of maximum, minimum temperature (˚C) and rainfall (mm).**

| Location | Statistics | Average maximum temperature (˚C) | | | | | Average minimum temperature (˚C) | | | | | Rainfall (mm) | | | | |
|---|---|---|---|---|---|---|---|---|---|---|---|---|---|---|---|---|
| | | Summer | Autumn | Winter | Spring | Annual | Summer | Autumn | Winter | Spring | Annual | Summer | Autumn | Winter | Spring | Annual |
| Hoedspruit (DS34) | Maximum | 34.8 | 34.6 | 29.0 | 33.0 | 30.2 | 23.0 | 21.2 | 15.4 | 20.35 | 17.8 | 422.5 | 175.7 | 82.8 | 167.9 | 988.1 |
| | Minimum | 28.1 | 24.6 | 22.7 | 25.5 | 27.3 | 14.7 | 10.1 | 6.5 | 12.0 | 14.3 | 2.8 | 0.0 | 0.0 | 0.0 | 176.9 |
| | Mean | 31.4 | 28.8 | 25.6 | 29.4 | 28.8 | 20.2 | 16.3 | 10.5 | 15.5 | 15.9 | 81.1 | 29.2 | 4.2 | 34.4 | 446.6 |
| | SD | 1.4 | 2.1 | 1.3 | 1.5 | 0.7 | 1.1 | 2.9 | 1.8 | 1.9 | 0.7 | 61.62 | 32.0 | 9.7 | 34.4 | 160.4 |
| | CV | 4.6 | 7.3 | 5.1 | 5.1 | 2.4 | 5.6 | 18.1 | 16.9 | 12.09 | 4.7 | 76.02 | 109.5 | 230.4 | 100.0 | 35.9 |
| | Z Test | +0.069 | +0.051 | +0.318 | +0.242 | +0.201 | -0.21 | -0.139 | +0.076 | -0.024 | -0.086 | -0.165 | -0.010 | -0.11 | -0.162 | -0.171 |
| | Sen's slope | +0.033 | +0.024 | +0.149 | +0.115 | +0.018 | -0.104 | -0.066 | +0.036 | -0.012 | -0.008 | -0.084 | -0.051 | -0.057 | -0.082 | -0.017 |
| | Trend | Increase | Increase | **Increase** | **Increase** | **Increase** | Decrease | Decrease | Increase | Decrease | Decrease | Decrease | Decrease | Decrease | Decrease | Decrease |
| | (p-value) Significance | 0.49 NS | 0.61 NS | **0.001 S** | **0.02 S** | **0.04 S** | **0.03 S** | 0.16 NS | 0.45 NS | 0.81 NS | 0.39 NS | 0.09 NS | 0.31 NS | 0.26 NS | 0.11 NS | 0.087 NS |
| Oudestad (DS63) | Maximum | 36.4 | 33.8 | 29.7 | 34.9 | 31.3 | 19.9 | 18.8 | 8.8 | 18.1 | 13.9 | 224.6 | 181.1 | 29.2 | 213.3 | 852.7 |
| | Minimum | 25.8 | 22.7 | 20.0 | 25.1 | 26.1 | 14.8 | 4.7 | 0.8 | 7.2 | 9.5 | 24.4 | 0.1 | 0 | 0.0 | 385.6 |
| | Mean | 31.0 | 27.7 | 23.43 | 29.46 | 27.9 | 17.6 | 12.1 | 4.7 | 13.8 | 11.9 | 101.7 | 38.4 | 4.4 | 62.13 | 620.0 |
| | SD | 1.6 | 2.4 | 1.8 | 1.8 | 1.1 | 0.9 | 3.6 | 1.6 | 2.5 | 0.7 | 39.9 | 34.83 | 6.9 | 48.2 | 96.1 |
| | CV | 5.2 | 8.7 | 7.9 | 6.1 | 4.0 | 5.3 | 30.2 | 34.7 | 17.9 | 5.7 | 39.3 | 90.72 | 156.6 | 77.5 | 15.5 |
| | Z Test | +0.46 | +0.27 | +0.596 | +0.645 | +0.53 | +0.062 | +0.071 | +0.171 | +0.032 | +0.18 | +0.067 | +0.026 | -0.12 | -0.038 | +0.066 |
| | Sen's slope | +0.24 | +0.14 | +0.303 | +0.328 | +0.055 | +0.032 | +0.036 | +0.087 | +0.017 | +0.018 | +0.034 | +0.013 | -0.062 | -0.019 | +0.007 |
| | Trend | **Increase** | Increase | Increase | Increase | Increase | Increase | Increase | Increase | Increase | Increase | Increase | Increase | Decrease | Decrease | Increase |
| | (p-value) Significance | **0.003 S** | **0.01 S** | **0.003 S** | **0.001 S** | **0.001 S** | 0.53 NS | 0.48 NS | 0.09 NS | 0.75 NS | 0.07 NS | 0.50 NS | 0.79 NS | 0.22 NS | 0.70 NS | 0.511 NS |

Significance: S- Significant, NS- Not Significant

rate of 0.055˚C per year at the 95% confidence level for the period under study. On the other hand, the mean annual Tmin varied between 9.5 to 13.9˚C with a mean value of 11.9 ± 0.7˚C (Table 4). The trend analysis of mean annual Tmin showed that it increased with a Z = 0.18 at the rate of 0.018˚C per year for the period of study. However, the rate of increase of mean annual Tmin was not significant at the 95% confidence level. Seasonally, both Tmin and Tmax show increasing trends across all seasons. The increasing trends are statistically significant for Tmax in all four seasons but statistically not significant for Tmin in all four seasons at the 95% confidence level. In general, mean daily Tmax and Tmin have increased by 0.42 and 0.27˚C respectively in the southern part of the Catchment.

The rainfall does not show a clear signal for an increasing or decreasing trend. A decreasing trend in total annual and seasonal rainfall in the northern (district 34) of the study area is indicated by the results. However, the trends are not statistically significant at the 95% confidence level. The mean total annual rainfall varied from 176.9 to 988.1 mm with a mean of 446.6 mm±160.4. In the southern (district 63) parts, the mean total annual rainfall varied from 385.6 to 852.7 mm with a mean of 620.0 mm±96.1 The Mann-Kendall trend analysis showed that test Z was +0.066 with a Sen's slope estimator of +0.007, which indicated that total annual rainfall has increased over the study period. In addition, the results indicate an increasing trend in total seasonal rainfall in summer. However, the trends are not significant at the 95% confidence level.

**Standardized Precipitation Index (SPI).** As shown in Fig 3 the SPI values suggest that the Catchment is characterized by drought conditions with extremely dry category (≤-2.0) occurring in 1992 and 2004 in both districts. The figure further suggests that drought has recently persisted over the Catchment particularly in the northern part (DS34) experiencing moderate to severe drought conditions. The results of the SPI affirm the historical wetter conditions of the southern part of the Catchment. The persistent drought conditions over the southern region can be deduced to be agricultural and hydrological droughts.

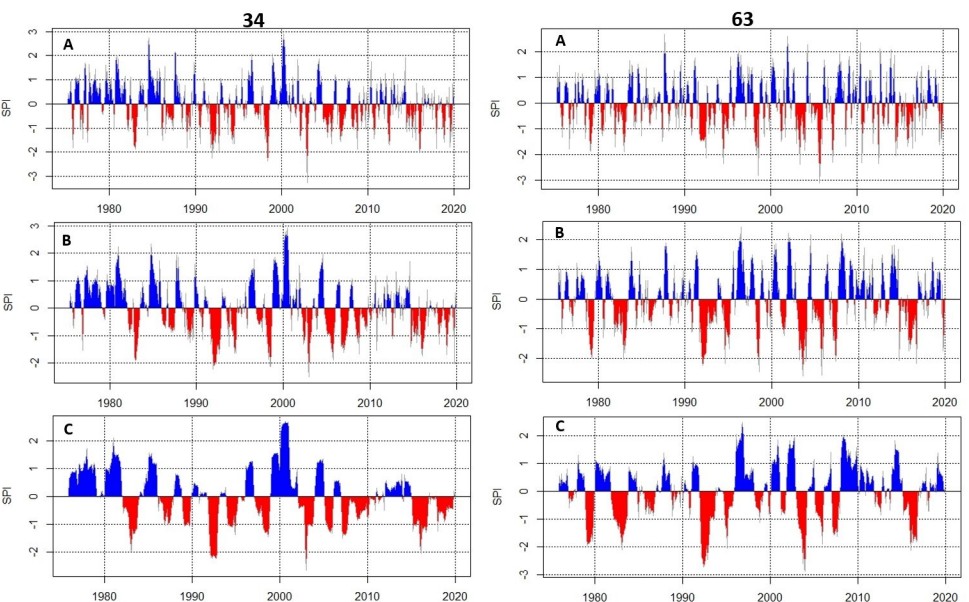

**Fig 3.** The Standardized Precipitation Index for (A) 2-, (B) 6-, and (C) 12-month accumulation period for two homogenous rainfall regions districts 34 and 63 of the Olifant River Catchment from 1976–2020.

## Model evaluation

The results of the effectiveness of the model to simulate historical temperatures and rainfall are shown in Fig 4 as an example for Hoedspruit and summarized in Table 6. Statistics for eight models were calculated with each model and the ensemble gave a colour code. The location of each colour on the plot indicates how correlated the model's simulated variables (rainfall and temperatures) match the observed pattern. The contours indicate the RMSE. The RMSE is the difference between the simulated and observed patterns which is proportional to the distance to the point on the x-axis, while the SD of the simulated pattern is proportional to the radial distance from the origin. The ensemble mean for Tmax has a pattern correlation with an observation of about 0.88. The RMSE between the simulated and observed patterns of maximum temperature is about 0.5°C/day while the SD of the simulated pattern for Tmax is about 0.8°C/day which less than the observed SD of 1.0°C/day. Overall, the CanESM2 and CSIRO models generally agree best with the observed maximum temperature, with both having RMSE within 0.65°C/day and 0.7°C/day respectively. However, the CSIRO model has a slightly higher correlation with observations and has the same standard deviation as the observed maximum temperature, whereas CanESM2 has an SD of 1.1°C/day compared to the observed value of 1.0°C/day.

On the other hand, the ensemble mean for Tmin gives a higher pattern correlation with an observation of about 0.97. The RMSE between the simulated and observed patterns of Tmin is about 0.2°C/day while the SD of the simulated pattern is about 0.9 compared to the observed SD of 1.0°C/day. All models present a high correlation with observation. The CanESM2 and CSIRO models generally agree best with Tmin observations, with both having RMSE within 0.38 and 0.37 respectively and the same standard deviation as the observed minimum temperature.

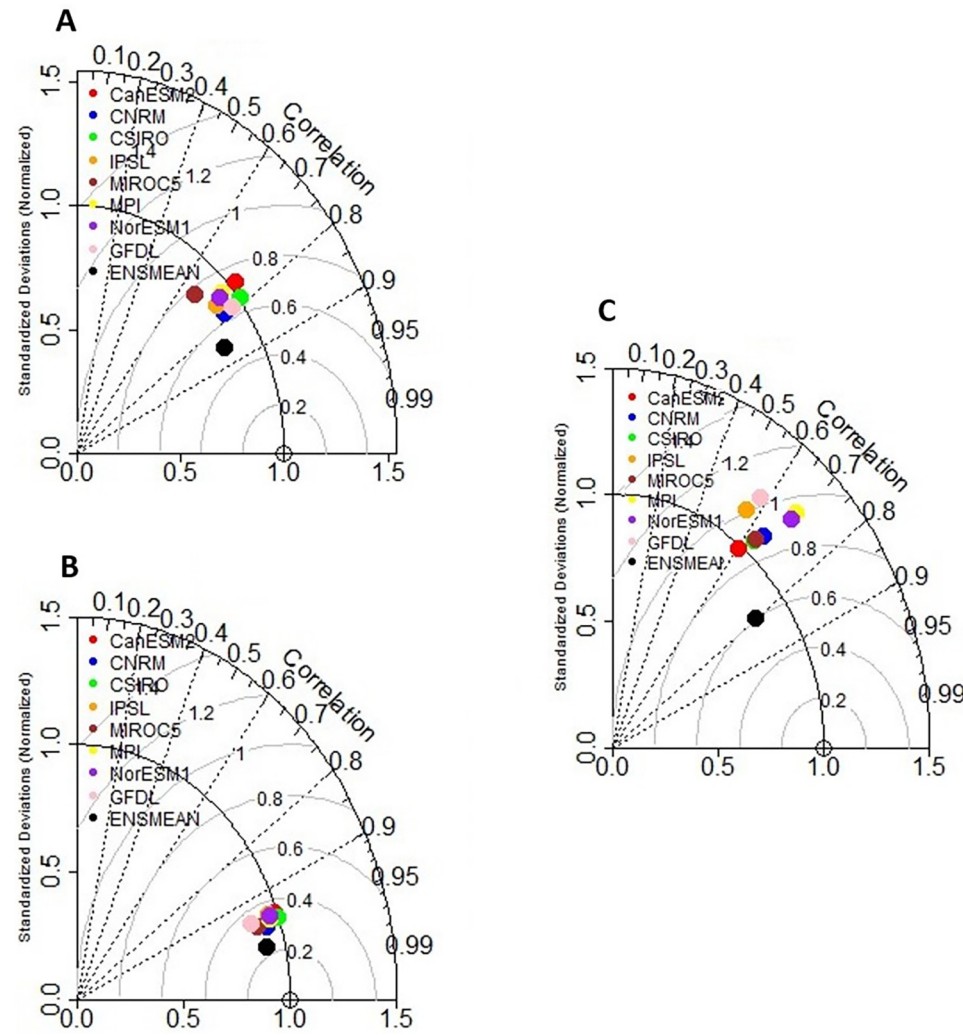

**Fig 4.** Taylor diagrams for (A) maximum temperature, (B) minimum temperature and (C) precipitation, comparing observations with CMIP5 models and ensemble mean simulations for Olifant River Catchment for the period 1976–2005.

While the ensemble mean of the member models produces a higher correlation of about 0.8 between the observed and model simulated annual total rainfall, there is largely less agreement between individual models and the observations. All models indicate a high RMSE and the spatial variability is higher with an average SD of 1.2 mm/day compared to the observed value of 1.0 mm/day except for the CanESM2 model. The Taylor diagram thus suggests that the ensemble models have a relatively high confidence level for temperature projections, whereas it has low confidence for precipitation projections.

**Table 6. Summary of Taylor diagrams for maximum, minimum temperature and precipitation, comparing observations with CMIP5 models and ensemble mean simulations for Olifant River Catchment for the period 1976–2005.**

| Variables | Correlation | Centred root-mean-square (RMS) | Standard deviation (SD) |
|---|---|---|---|
| Maximum Temperature | 0.88 | 0.5 | 0.8 |
| Minimum Temperature | 0.97 | 0.2 | 0.9 |
| Rainfall | 0.80 | 0.6 mm/day | 0.75 mm/day |

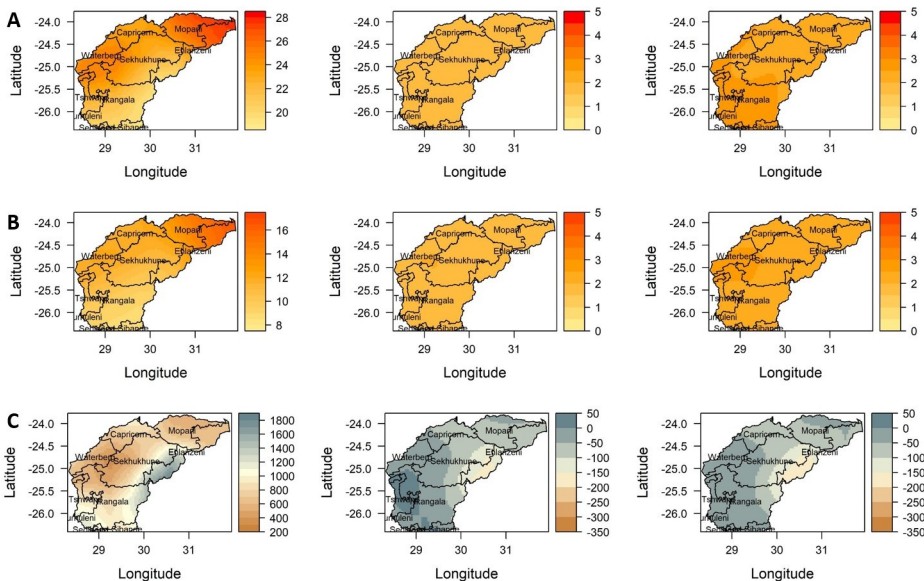

**Fig 5.** Projected changes in annual (A) mean maximum, (B) minimum temperature and (C) total rainfall, for 2036–2065 (2nd column) and 2066–2095 (3rd column) periods under scenarios of the RCP4.5 relative to the baseline period 1976–2005 (1st column) over the Olifants River Catchment. "Basemap (Water management shapefile was collected from the South Africa National Department of Water and Sanitation on http://www.dwa.gov.za/iwqs/gis_data/" [29].

**Climate change projections for Olifants River Catchment.** Projected changes in the annual values of total rainfall, mean Tmax and Tmin over the Olifants River Catchment for the periods 2036–2065 (near future) and 2066–2095 (far future) under RCP 4.5 and RCP8.5 scenarios, relative to the reference period 1976–2005 are presented in Figs 5 and 6 respectively, with a summary in Table 7. At mid-century 2050 (2036–2065), under the RCP4.5 condition, Tmax is projected to increase between a minimum of about 1.8 and a maximum of 2.0˚C. In

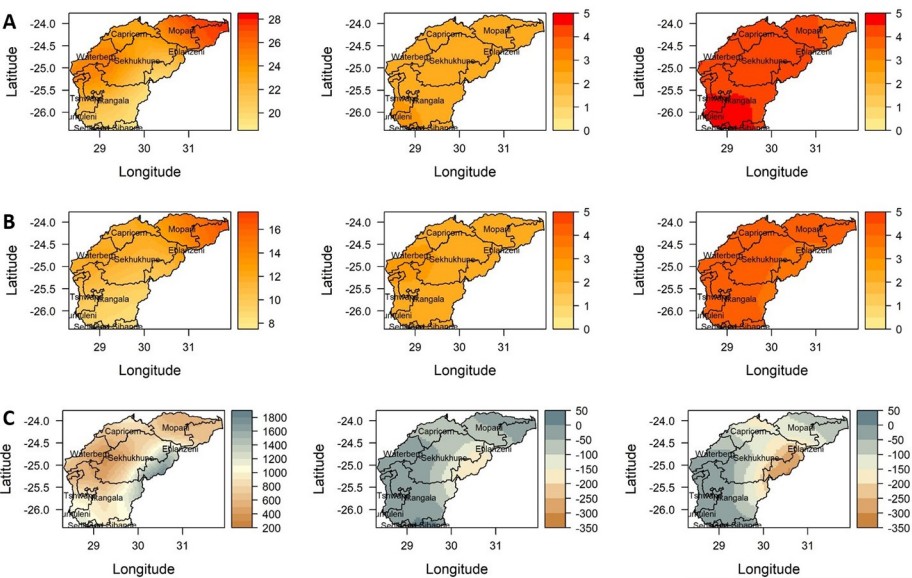

**Fig 6.** Projected changes in annual (A) mean maximum, (B) minimum temperature and (C) total rainfall, for 2036–2065 (2nd column) and 2066–2095 (3rd column) periods under scenarios of the RCP8.5 relative to the baseline period 1976–2005 (1st column) over the Olifants River Catchment. "Basemap (Water management shapefile was collected from the South Africa National Department of Water and Sanitation on http://www.dwa.gov.za/iwqs/gis_data/" [29].

**Table 7. Summary of historical and projected changes in mean maximum temperature, minimum temperature, and total rainfall for 2036–2065 and 2066–2095 periods under RCP4.5 and RCP8.5 scenarios relative to the baseline period 1976–2005 over the Olifants River Catchment.**

| Variables/Statistics | RCP4.5 | | | | | |
|---|---|---|---|---|---|---|
| | 1976–2005 | | 2036–2065 (change) | | 2066–2095 (change) | |
| | Min | Max | Min | Max | Min | Max |
| Maximum Temperature (˚C) | 19.2 | 27.9 | 1.8 | 2.0 | 2.2 | 2.6 |
| Minimum Temperature (˚C) | 8.5 | 16.6 | 1.7 | 2.0 | 2.1 | 2.6 |
| Rainfall (mm) | 336.19 | 1784.57 | -230.8 | 30.3 | -350.0 | 20.0 |
| RCP8.5 | | | | | | |
| Maximum Temperature (˚C) | 19.2 | 27.9 | 2.2 | 2.5 | 3.9 | 4.6 |
| Minimum Temperature (˚C) | 8.5 | 16.6 | 2.1 | 2.6 | 3.7 | 4.5 |
| Rainfall (mm) | 336.2 | 1784.6 | -297.0 | 20.3 | -429.8 | 10.9 |

addition, Tmax is likely to increase by a minimum of 2.2 and 2.6˚C under the RCP4.5 condition by the end of the century (Fig 5).

Under the RCP8.5 scenario, shown in Fig 6 and summarized in Table 7, Tmax is predicted to increase by a minimum of about 2.2 and a maximum of 2.5˚C by mid-century and is further projected to increase between a minimum of 3.9 and 4.6˚C by the end of the century. In general, Tmax is averagely projected to be warmer by about 2.2˚C in the northern part (Mopani, Ehlanzeni districts) and by 2.5˚C in the southern part (Ekurhuleni, Gert Sibande, Sedibeng and City of Tshwane districts) of the catchment under RCP4.5 and to increase by about 4.3˚C in the southern part and 3.8˚C in the northern part under RCP8.5 compared to the baseline period of 1976–2005. On the other hand, Tmin is projected to increase between 1.7–2.0˚C and between 2.1–2.6˚C under RCP4.5 by mid-century and end of the century respectively. Higher Tmin values are projected under the RCP8.5 scenario with an increase between 2.1–2.6˚C and 3.7–4.5˚C by mid-century and end of the century respectively. Averagely, Tmin is projected to increase by about 1.6˚C in the northern and central parts, and by about 2.0˚C in the southwest including the Waterberg district under the RCP4.5 scenario and estimated to increase by about 3.8˚C in the southwestern part and by about 3.6˚C in the northern and central parts under the RCP8.5 scenario, relative to the baseline period of 1976–2005 (Fig 6). Generally, the projected changes in temperature and rainfall vary substantially across the catchment. These variations can be attributed to the complexity of the landscape and the associated climate fluctuations as shown in the historical trend analysis.

It is important to note that there is high uncertainty in the direction and amount of change in total annual rainfall across the ensemble models. The projection suggests a decrease in total annual rainfall by about 5–30%, especially over the eastern to central parts of the catchment by mid-2050 under both RCP4.5 and 8.5 scenarios. Regions with less rainfall historically are projected to have marginal increases in rainfall. The southern region of the catchment is projected to receive about a 20mm p.a. (5%) increase in rainfall. The results indicate a further decrease in rainfall (about 10–40%) over a large portion of the catchment going into the far future. The central interior part of the catchment is projected to experience a significant decrease in annual rainfall up to about 25%, with increased drying over time. Despite predictions of general drying conditions over most of the catchment, slight to moderate rainfall increases are projected over the south-western parts of Olifants River Catchment in spring and summer, although statistically not significant.

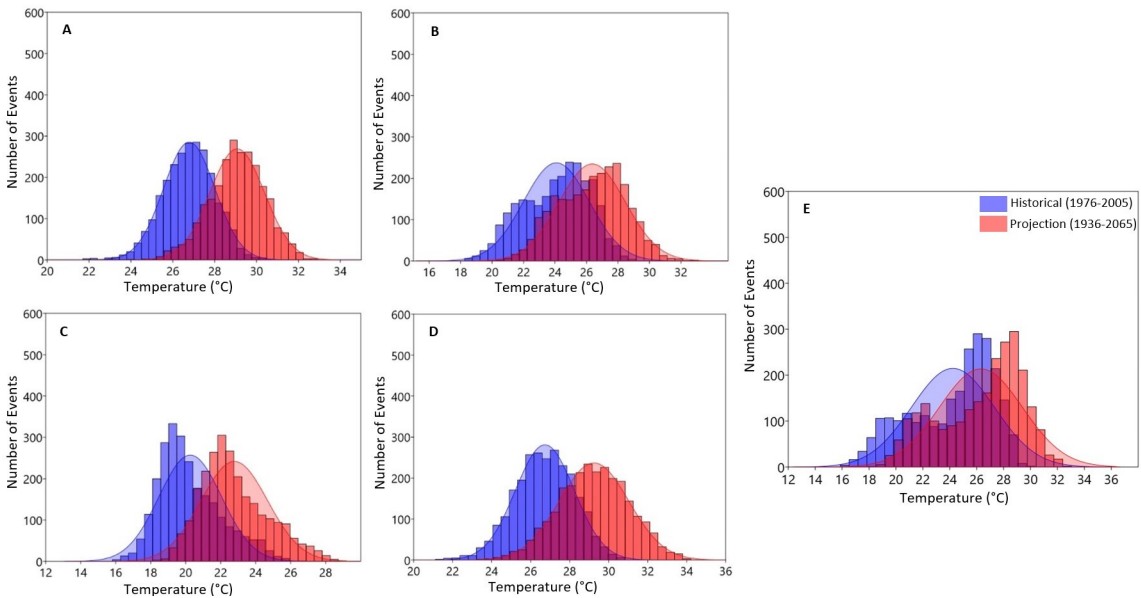

**Fig 7.** Probability distribution estimation of daily seasonal A) Summer, B) Autumn, C) Winter, D) Spring and E) annual maximum temperatures Over Olifants Catchment (1976–2005 vs. 2036–2065).

The results of the occurrence of both the historical and projected Tmax, Tmin and rainfall generated using the probability distribution function over the entire catchment are shown in Figs 7–11. The figures present the averages of the variables as well as the lower and upper extreme boundaries of the projected climate. As shown in Fig 7, the number of days (events) with an annual maximum temperature > 32˚C is projected to increase by 2036–2065 (Fig 7E).

Seasonally, mean Tmax is expected to increase above 31˚C in summer, 29˚C in autumn, 26˚C in winter and 31˚C in spring by the mid-2050s (Fig 7A–7D respectively). These maximum temperature values are projected to further increase in the far future reaching above 33˚C in summer, 31˚C in autumn, 26˚C in winter and 32˚C in spring (Fig 8A–8D respectively) with an increasing number of occurrences.

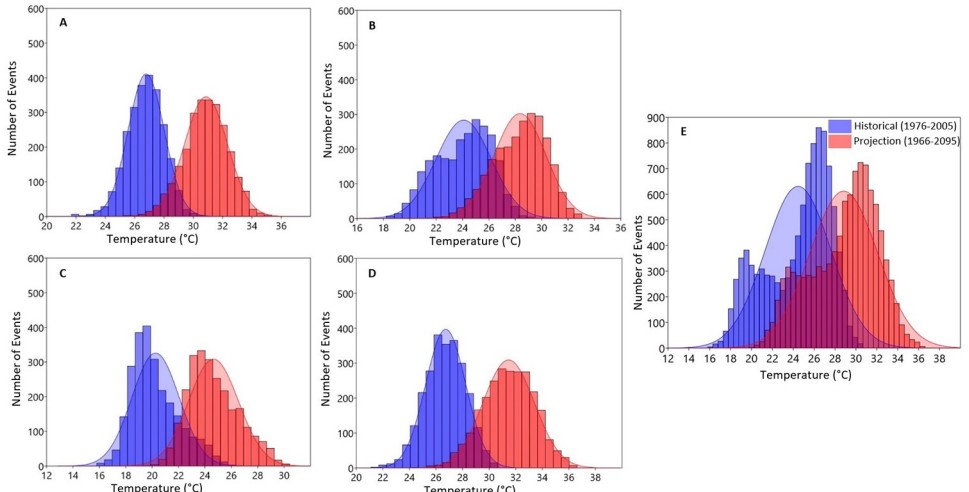

**Fig 8.** Probability distribution estimation of daily seasonal A) Summer, B) Autumn, C) Winter, D) Spring and E) annual maximum temperatures Over Olifants Catchment (1976–2005 vs. 2066–2095).

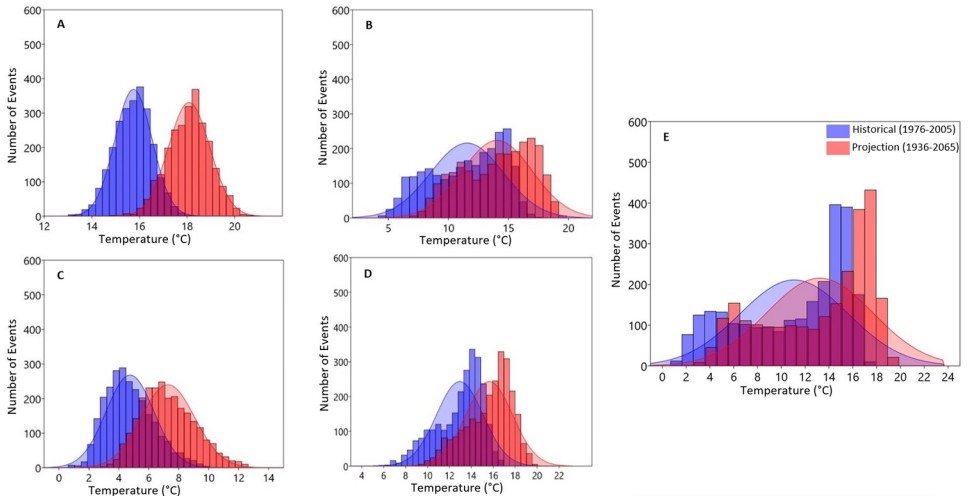

**Fig 9.** Probability distribution estimation of daily seasonal A) Summer, B) Autumn, C) Winter, D) Spring and E) annual minimum temperatures Over Olifants Catchment (1976–2005 vs. 2036–2065).

On the other hand, Tmin > 17˚C is projected to increase over the catchment (Fig 9E) in the mid-2050s. Seasonally, the number of events of Tmin > 18˚C is projected to increase in the summer months. In general, all seasons are projected to experience higher Tmin under the RCP8.5 scenario by the mid-2050s (Fig 9A–9D respectively).

By the mid-2080s minimum temperature values are further projected to increase above 19˚C with increased days of occurrence in summer (Fig 10A). Average Tmin values are projected to increase above 17˚C, 11˚C and 16˚C in autumn, winter and spring respectively (Fig 9B–9D).

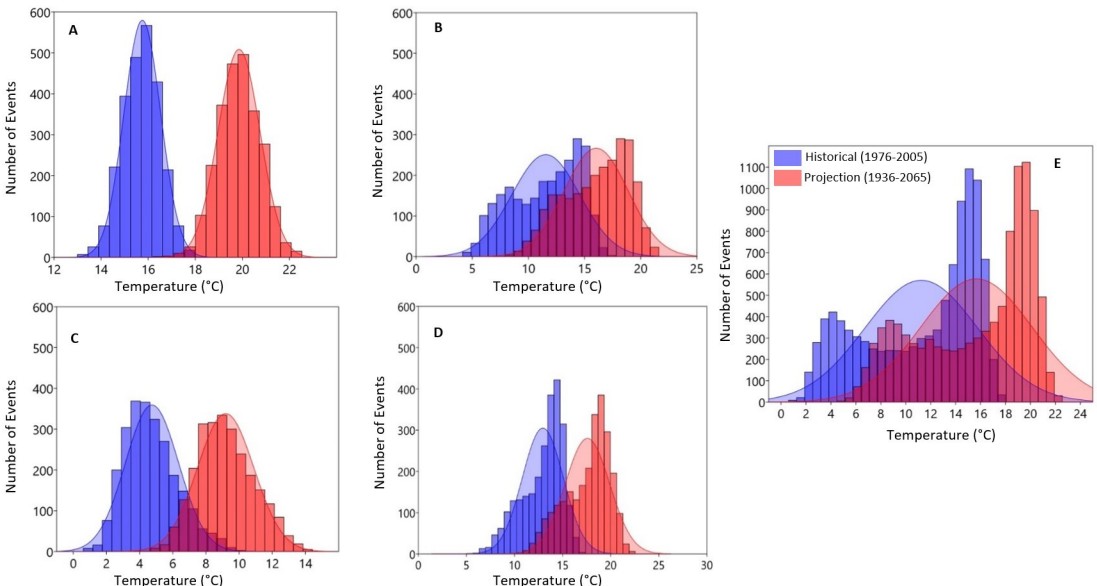

**Fig 10.** Probability distribution estimation of daily seasonal A) Summer, B) Autumn, C) Winter, D) Spring and E) annual minimum temperatures Over Olifants Catchment (1976–2005 vs. 2066–2095).

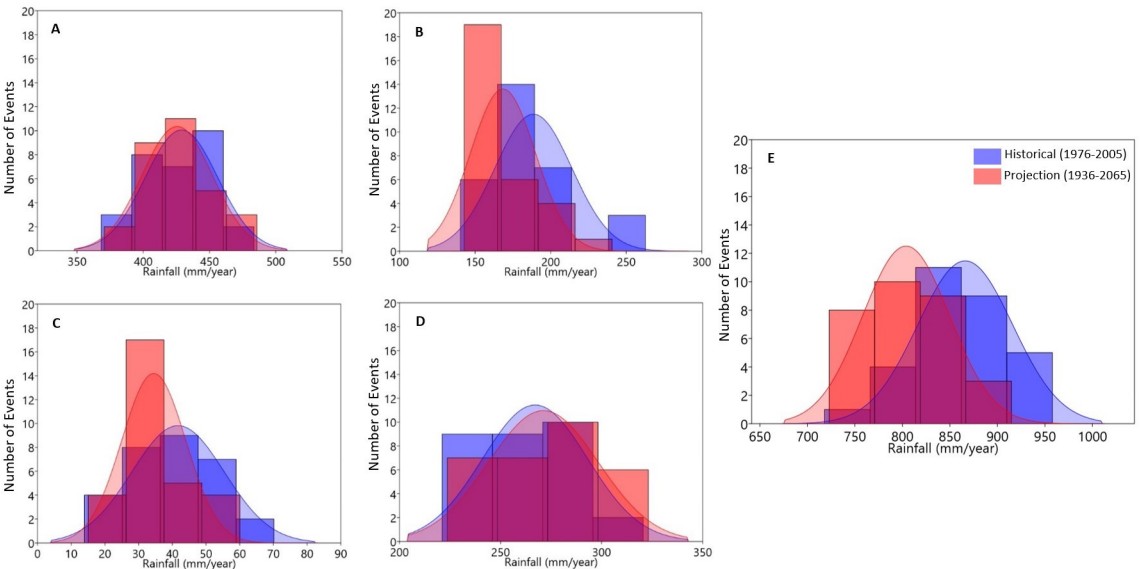

**Fig 11. Probability distribution estimation of seasonal and annual total precipitation over the Olifants catchment (1976–2005 vs. 2026–2055) with relevant gamma distributions superimposed.**

Fig 11 presents the histograms of annual and seasonal monthly total rainfall for the historical period (1976–2005) and projected (2036–2065) indicating the expected changes in the annual and seasonal rainfall distributions over the Olifants River Catchment, as predicted by the CORDEX multi-model mean. The Fig shows that annual total rainfall is projected to reduce in amount and the number of rainfall events (Fig 11E). The declining trend in total rainfall is the same across the season except for a projected increase in the events of rainfall amount > 450 mm in summer and > 300 mm in spring.

## Discussion

The historical analysis of available temperature and rainfall time series in the Olifants River Catchment indicates increasing trends in temperature and a generally decreasing trend in rainfall. Specifically, climate projections anticipate that temperature values will further increase, and rainfall decrease under both the RCP4.5 and RCP8.5 scenarios. The results indicate an increasing trend in historical and projected Tmax and Tmin between 1–5°C, particularly in the southern part of the catchment and a decreasing trend in both historical and future rainfall of about 6–35% across the catchment as a whole. Specifically, a major decline in rainfall is projected over the central part of the catchment.

### General trends in temperature and rainfall distribution patterns

Regarding the mean surface temperature, historically, the northern parts of the catchment, including the sections of the Mopani and Ehlanzeni districts of the Catchment, are warmer while the southern parts include sections of Ekurhuleni, Gert Sibande, Sedibeng and City of Tshwane districts are relatively cooler. However, the projected climate change under the two scenarios of RCP 4.5 and 8.5, suggests that the southern parts of the catchment are likely to get warmer than the northern parts in general. The findings of this study agree with the findings of [49] who reported significant increasing trends in warm temperatures over individual stations located in the northern sections of South Africa.

A change in the general rainfall distribution patterns is also expected. The central region of the catchment that includes the sections of the eastern part of Sekhukhune and the northern part of Ehlanzeni districts with historically relatively higher rainfall is projected to experience a greater reduction of rainfall received, at about 30%. In general, the results indicate that the changes will be more apparent after about mid-century. In addition, the SPI analysis affirms the persistence of frequent drought conditions over the Catchment, particularly over the southern region which recently experienced drought. The results of the decreasing trends in observed rainfall are consistent with the individual station analysis conducted by [33], which found long-term increases in rainfall in the southern interior and decreases in the far north-east, but no significant trends in annual and seasonal rainfall over most of the remainder of South Africa.

The spatial variability in rainfall can be due to varying weather systems that characterize the study area. Rainfall District 34 falls in the Lowveld, and although drying is evident in the Mopani district portion of the catchment, the rainfall statistics show no drying trend in the Lowveld. The Lowveld is very dependent on moisture from the ridging of the Indian Ocean High, and with the expected strengthening of the subtropical high-pressure belt, the influx of moist air from the east will probably not diminish.

## Model simulations vs. observations

Although there is a moderate correlation between simulated and observed rainfall, climate models are still useful as they are largely designed to predict the trend in climatic change over time rather than to predict specific weather events. However, we suggest that interpretation and usage of the results of rainfall trends should be guided by historical understanding. However, the results show a strong correlation between simulated and observed temperatures. Hence, the utility of temperature projections is more reliable, especially if used to estimate the rate of change in temperature instead of absolute values.

## Probable impacts of climate change

Given the historical increasing drying and a warming trend in rainfall and temperatures over the catchment respectively, climate change is expected to increase the probability of shortages in water supply with both direct and indirect impacts on water-dependent sectors. As temperatures rise and with possible reductions in rainfall, crop yield could be affected, and food security could be compromised. Specifically, the prediction points to a likely scenario of increasing temperatures together with more dry periods which in turn points to an increased likelihood of severe droughts. Particularly, farmers who rely heavily on rain-fed agriculture will be the most adversely affected by climate change. Obviously, the projected climate changes in the catchment will have varying impacts on various sectors. For example, human health could be impacted by an increased incidence of malaria, cholera, and diarrhoea [50], with an indirect consequent negative impact on tourism. The probable impacts should take into account the demographical and socio-economic situation in the catchment, both current and projected, as in the approach in the Green Book [51].

## CMIP 6 projections

Since the CMIP6 results have become available it will be useful to make a brief comparison between the results in the study and the latest trend projections [52]. The CMIP6 used the Shared Socialeconomic Pathways (SSPs) as against the RCPs in CMIP5. The CMIP6 projection of the average temperature increase for the Catchment at SSP3-4.5 is an increase of approximately 3°C from the base period of 1961 to 1990 to the projected period of 2081–2100. At

SSP5-8.5 the increase is projected at 5–6˚C (https://interactive-atlas.ipcc.ch/). Therefore, the CMIP6 temperature projections are within range of the projections derived in this study. The CMIP6 rainfall projections at both SSP3-4.5 and SSP5-8.5 indicate a non-significant 4–6% decrease in rainfall, which is close to the previous CMIP5 projections. These projections are in contrast with the results of this study which indicate reductions of up to 35%, illustrating the need for downscaling of the CMIP6 projections and the interrogation of individual ensemble members to obtain a clearer idea of probable worst-case scenarios to be expected at catchment-scale.

## Conclusion and recommendations

Climate-induced threats are anticipated to increase in duration, frequency, and severity under climate change. This study examined trends in historical annual and seasonal temperatures and rainfall and investigated future projections over the Olifants River Catchment to understand the region's vulnerability to climate change. Daily values of minimum and maximum temperature from two ground weather stations and corresponding SAWS' district of homogenous monthly rainfall data were used. An ensemble of eight global climate model simulations of the CORDEX Africa forced with CMIP5 models was used to make future climate change projections under the RCP4.5 and RCP 8.5 climate scenarios for two time periods of 2036–2065 (near future) and 2066–2095 (far future). The ensemble of the RCMs is well able to depict the spatial distribution of temperatures and rainfall when compared with historical records indicating that it outperformed the individual models. However, uncertainties are associated with rainfall direction and the amount of change. While CMIP5 and CMIP6 results are comparable, the need for updated RCM outputs based on CMIP6 and the analysis of outputs of individual ensemble members are needed, considering the larger reductions in rainfall at the higher-resolution regional scale compared to the average of the GCM outputs in the same region.

The findings of these analyses are essential for developing adaptation strategies for the various economic sectors functioning in the Olifants River Catchment. In particular, the Ehlanzeni, Gert Sibande, and Sedibeng portions of the catchment which are projected to have higher values of increasing temperatures and decreasing rainfall will need the development of effective climate change adaptation tools. With vast economic activities such as mining and agriculture in the catchment largely dependent on water resources (rainfall), the results of this research are important for developing an impact assessment and adaptation strategies to ensure that the region continues to contribute to the national GPD.

We suggest that the government through the relevant structure should facilitate the provision and access to simplified information on anticipated changes in climatic variables and plausible impacts for varying climate-sensitive sectors, particularly agriculture and water. In addition, the provision of essential tools for decision making for present and future management and practices among direct users such as farmers are imperative for climate change adaptation purposes. Furthermore, given the results, the achievement of various sustainable development goals such as Good Health and Well-being, Clean Water and Sanitation, Zero Hunger, Decent Work and Economic Growth, and No Poverty are likely to be further challenged by climate change in South Africa.

## Acknowledgments

The authors present their warm thanks to South32 for inspiring the research through a climate change assessment consultation.

## Author Contributions

**Conceptualization:** Abiodun Morakinyo Adeola.

**Data curation:** Abiodun Morakinyo Adeola, Andries Kruger.

**Formal analysis:** Abiodun Morakinyo Adeola, Andries Kruger, Thabo Elias Makgoale, Joel Ondego Botai.

**Methodology:** Abiodun Morakinyo Adeola.

**Writing – original draft:** Abiodun Morakinyo Adeola, Andries Kruger.

**Writing – review & editing:** Abiodun Morakinyo Adeola, Andries Kruger, Thabo Elias Makgoale, Joel Ondego Botai.

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
