## [Decision Letter · Decision Letter 0]

2 Dec 2021

PONE-D-21-31259Trends and projections of surface temperature and rainfall over the Olifants River catchmentPLOS ONE

Dear Dr. Adeola,

Thank you for submitting your manuscript to PLOS ONE. After careful consideration, we feel that it has merit but does not fully meet PLOS ONE’s publication criteria as it currently stands. Therefore, we invite you to submit a revised version of the manuscript that addresses the points raised during the review process.

The authors have evaluated the trends of surface temperature and rainfall over the Olifants River catchment, which is interesting and important to understand the current state of climate changes at basin scale. However, there are some major issues that pointed by all the four reviewers need to be clarified and resolved by the authors. Thank you. 

We look forward to receiving your revised manuscript.

Kind regards,

Mou Leong Tan

Academic Editor

PLOS ONE

Journal Requirements:

2. Please amend your Methods section to provide a URL where the SAWS data base can be accessed.

 [Partial funding for the research was received from the South32 mining company].

5. We note that Figures 1 and 5 in your submission contain map images which may be copyrighted. All PLOS content is published under the Creative Commons Attribution License (CC BY 4.0), which means that the manuscript, images, and Supporting Information files will be freely available online, and any third party is permitted to access, download, copy, distribute, and use these materials in any way, even commercially, with proper attribution. For these reasons, we cannot publish previously copyrighted maps or satellite images created using proprietary data, such as Google software (Google Maps, Street View, and Earth). For more information, see our copyright guidelines: http://journals.plos.org/plosone/s/licenses-and-copyright.

a) You may seek permission from the original copyright holder of Figures 1 and 5 to publish the content specifically under the CC BY 4.0 license.  

6. We noticed you have some minor occurrence of overlapping text with the following previous publication(s), which needs to be addressed:

- https://link.springer.com/article/10.1007%2Fs00704-019-03074-6

- https://www.sciencedirect.com/science/article/pii/S0048969718344504?via%3Dihub

In your revision ensure you cite all your sources (including your own works), and quote or rephrase any duplicated text outside the methods section. Further consideration is dependent on these concerns being addressed.

Reviewers' comments:

Reviewer's Responses to Questions

**Comments to the Author**

1. Is the manuscript technically sound, and do the data support the conclusions?

Reviewer #1: Partly

Reviewer #2: Partly

Reviewer #3: No

Reviewer #4: Partly

2. Has the statistical analysis been performed appropriately and rigorously? 

Reviewer #1: Yes

Reviewer #2: Yes

Reviewer #3: No

Reviewer #4: Yes

3. Have the authors made all data underlying the findings in their manuscript fully available?

Reviewer #1: Yes

Reviewer #2: Yes

Reviewer #3: Yes

Reviewer #4: Yes

4. Is the manuscript presented in an intelligible fashion and written in standard English?

Reviewer #1: No

Reviewer #2: No

Reviewer #3: No

Reviewer #4: No

5. Review Comments to the Author

Reviewer #1: Interesting study on the observed trends and future projections for temperature and precipitation in the Olifants River, South Africa. Such regional analyses are missing from the literature. Although this is a good effort and the study could contribute to a better understanding of regional climate change impacts, I have several concerns regarding the datasets and depth of analysis, particularly for future projections.

Major points:

- The 50-km CORDEX-AFRICA projections are not state-of-the-art at the moment. I strongly recommend using the CORDEX-CORE data that are available in a 25-km spatial resolution. Although fewer models are available in the CORDEX-CORE ensemble, the increased resolution could add value.

- The Taylor diagrams alone do not provide a complete view of model biases. A summary table with descriptive statistics, like the one used for observations, would be more useful.

- I strongly recommend exploring also a more optimistic pathway (i.e., RCP2.6 or RCP4.5) for providing a more complete picture of the expected ranges of change.

- Although there has been sufficient statistical analysis of observed conditions, the robustness and significance of future projections are not discussed at all.

- I recommend changing the title to: “Observed trends and projections of temperature and precipitation in the Olifant’s river catchment in South Africa”.

Reviewer #2: The authors used the historic observation data and future model outputs to study trends of temperature and rainfall in Olifants River Catchment. The study is interesting but there still exits strong weakness in as follows:

Majors:

1. The title is too general and should give the specific scientific meaning.

2. For the temperature trend of page 10, " However, unlike the maximum temperature, the rate of decrease of mean annual minimum temperature was not significant at the 95% confidence level.” What mechanism for it not passing the statistical test?

3. In page 11, “The rainfall does not show a clear signal for increasing or decreasing trend. A decreasing trend in total annual and seasonal rainfall in the northern (district 34) of the study area is indicated by the results. However, the trends are not statistically significant at the 95% confidence level.” What reason for the test? The authors can not only describe the result, and the physical mechanism behind it MUST be given for a scientific paper.

Minors:

1. Line 3 of page 2 in the introduction: The sentence "and development challenges in the " is not finished.

2. Line 1-3 of page 3: "while the link between increased temperatures and changes in rainfall has been modelled in detail, the same is not true for the effect on specific river catchments and the recharge of underground waters. " The description is not clear, what link as mentioned?

4. Line 5 of paragraph 3 of page 3: "Studies have shown that GCMs are able in reproducing temperature trends realistically" is suggested to change as "Studies have shown that GCMs are able to reproduce temperature trends realistically".

5. Line 8 of paragraph 3 of page 3: For the description "require a deeper knowledge of the local conditions", the verb should be "requires".

6. Line 1 of page 7 below the equation: “annual or seasonal values” has not the same font size as other words. The same problem appears to be below equation 2.

7. The table 4 of page 12： The word “Increase ” appears in two lines many times.

8. Line 10 of page 13: "there still exist a weak correlation" is error in grammar. And yhy line numbers begin this page and not in page 1-12?

9. Line 38-39 of page 14: " Areas with historically less rainfall is projected to ......" is error in grammar.

10. Line 86-87 of page 16: "It is anticipated that future temperature will increase, and rainfall decrease under the business-as- usual RCP8.5 scenario over the Olifants River Catchment." The description as “future temperature will increase in RCP8.5” does not make any sense in science.

11. Line 87-88 of page 16: "The findings of this study as related to temperature agrees with the findings of " is wrong in grammar.

12. Line 102-103 of page 16: "As temperatures rise and rainfall decreases, crop yield will be affected" is wrong in grammar.

13. Line 118 of page 17: "An ensemble of eight regional climate model simulations of the CORDEX Africa forced with CMIP5 models " What are the eight regional climate model?

14. The reference in page 19 should be list in the same format. The discrepancy is presented like "(2019)" of citation 2 and “2006”in 3. Please check all reference.

Reviewer #3: Require major revision , please see comments in attachment. This is a potentially very useful paper, but the current version and some work with the discussions and additional inputs in the methods and objectives.

The methods and results are interesting, but the presentation of the manuscript is poor, particularly at the discussion level. Based on the discussion section, I think the manuscript is now more written as an internal report, and not a qualitative research paper. This study lacks a discussion section that in quantitative explaining the observed results. The conclusion actually is longer than the discussion not unless I am missing out on the whole context. This is quite a shame because the writing is really well done with logical flow with this information surely very useful in the advancement of the climate prediction agenda based on downscaled models.

There is no clear objective and specific objectives, research question or hypothesis formulated. Instead, the authors need to come up with a clear research question, rethink the statistics they use on the data based on that question, elaborate in the methods section, and then write a focused discussion.

Reviewer #4: The authors attempt to predict the future trends and projections of surface temperature and rainfall over the Olifants River catchment using downscaling. The manuscript lacks coherent structure and suffers from methodological gaps and unfounded conclusions. The paper has a medium quality in terms of the language, with various syntax mistakes (extra checks are required). The introduction contains a lot of information, but fails to provide the background intelligible and outline the overall scope of the study adequately. When it comes to the description of the methodology, many confusions arose in the ‘data set’ portion as it was found difficult to identify which data set was used for the present study. Please add further explanation for the experimental design (in the methods section). In addition, when the readers reach the results and conclusions section, poor presentation (in terms of image resolution and position) of Figures made it difficult to identify the findings. Summarily, this manuscript seems to be a simple effort by the authors to present future climates projected by CMIP5 models. Based on the current state of the manuscript, I suggest that the research should only be accepted for publication if the authors complete the major revisions suggested in the uploaded manuscript.

Since, line number is not given properly; I did my corrections in the submitted PDF. Kindly go through it and do the necessary corrections.

6. PLOS authors have the option to publish the peer review history of their article (what does this mean?). If published, this will include your full peer review and any attached files.

Reviewer #1: No

Reviewer #2: No

Reviewer #3: **Yes: **Godwin Leslie muhati

Reviewer #4: No

---

## [Author Response · Author response to Decision Letter 0]

25 Jan 2022

Reviewer #1:

Interesting study on the observed trends and future projections for temperature and precipitation in the Olifants River, South Africa. Such regional analyses are missing from the literature. Although this is a good effort and the study could contribute to a better understanding of regional climate change impacts, I have several concerns regarding the datasets and depth of analysis, particularly for future projections.

Response: 

Thank you for the positive feedback. We appreciate the time and effort that the Reviewer has invested for understanding of our work and for the valuable comments for improvement of our manuscript. We appreciate the suggestions which help to improve the readability of our manuscript.

Responses to Reviewer #1 comments.

Comment: 

The 50-km CORDEX-AFRICA projections are not state-of-the-art at the moment. I strongly recommend using the CORDEX-CORE data that are available in a 25-km spatial resolution. Although fewer models are available in the CORDEX-CORE ensemble, the increased resolution could add value.

Response: 

The authors strongly agree with the reviewer’s comment. The authors acknowledge the value added of the CORDEX-CORE given its double resolution advantage over the CORDEX-AFRICA. The authors are of the opinion that the CORDEX-AFRICA has not been extensively used over the Southern Africa as compared to other IPCC regions. Also, we believe that this current study can form a bases for a value add to a further analysis of CORDEX-CORE. This study can be followed up with a comparison analysis between the 50-km and 25-km data. Hence, the authors proposed a future research using the CORDEX-CORE and make a comparison of its performance against the CORDEX-AFRICA while we hope that additional models will be made available in the CORDEX-CORE.

Comment: 

The Taylor diagrams alone do not provide a complete view of model biases. A summary table with descriptive statistics, like the one used for observations, would be more useful.

Response: 

A summary table has been provided as requested by the reviewer.

Comment: 

I strongly recommend exploring also a more optimistic pathway (i.e., RCP2.6 or RCP4.5) for providing a more complete picture of the expected ranges of change.

Response: 

Authors have included the RCP4.5 in the revised version of the manuscript

Comment: 

Although there has been sufficient statistical analysis of observed conditions, the robustness and significance of future projections are not discussed at all.

Response: 

We have now discussed the significance of future projections on the Catchment

 Comment: 

I recommend changing the title to: “Observed trends and projections of temperature and precipitation in the Olifant’s river catchment in South Africa”.

Response: 

The title has been changed accordingly.

Thank you again for the positive feedback and thank you for your consideration. We look forward to receiving a good review of this paper. 

Reviewer #2:

Comment: 

The authors used the historic observation data and future model outputs to study trends of temperature and rainfall in Olifants River Catchment. The study is interesting but there still exits strong weakness in as follows:

Response: 

Thank you for the positive feedback. We appreciate the time and effort that the Reviewer has invested for understanding of our work and for the valuable comments for improvement of our manuscript. We appreciate the suggestions which help to improve the readability of our manuscript.

Comment: 

The title is too general and should give the specific scientific meaning.

Response: 

The title has been changed has recommended.

Comment: 

For the temperature trend of page 10, " However, unlike the maximum temperature, the rate of decrease of mean annual minimum temperature was not significant at the 95% confidence level.” What mechanism for it not passing the statistical test?

Response: 

This is as indicated by the statistical trend analysis. This result could be due to the fact that District 34 falls in the Lowveld, and although drying is evident in northern Limpopo the rainfall statistics show no drying trend in the Lowveld. https://www.weathersa.co.za/home/extremeclimateindices The Lowveld is very dependent on moisture form the ridging of the Indian Ocean High, and even with the strengthening of the High Pressure belt, influx of moist are probably not start to diminish. This has been incorporated into the revised version of the manuscript.

Comment: 

In page 11, “The rainfall does not show a clear signal for increasing or decreasing trend. A decreasing trend in total annual and seasonal rainfall in the northern (district 34) of the study area is indicated by the results. However, the trends are not statistically significant at the 95% confidence level.” What reason for the test? The authors can not only describe the result, and the physical mechanism behind it MUST be given for a scientific paper.

Response: 

This result could be due to the fact that District 34 falls in the Lowveld, and although drying is evident in northern Limpopo the rainfall statistics show no drying trend in the Lowveld. https://www.weathersa.co.za/home/extremeclimateindices The Lowveld is very dependent on moisture form the ridging of the Indian Ocean High, and even with the strengthening of the High Pressure belt, influx of moist are probably not start to diminish. This has been incorporated into the revised version of the manuscript.

Minors:

Comment: 

Line 3 of page 2 in the introduction: The sentence "and development challenges in the " is not finished.:

Response: 

Comment has been completed.

Comment: 

Line 1-3 of page 3: "while the link between increased temperatures and changes in rainfall has been modelled in detail, the same is not true for the effect on specific river catchments and the recharge of underground waters. " The description is not clear, what link as mentioned?

Response: 

The readability of the sentence has been improved and the whole manuscript.

Comment: 

Line 5 of paragraph 3 of page 3: "Studies have shown that GCMs are able in reproducing temperature trends realistically" is suggested to change as "Studies have shown that GCMs are able to reproduce temperature trends realistically".

Response: 

Comment has been implemented accordingly.

Comment: 

Line 8 of paragraph 3 of page 3: For the description "require a deeper knowledge of the local conditions", the verb should be "requires".

Response: 

Comment has been implemented accordingly.

Comment: 

Line 1 of page 7 below the equation: “annual or seasonal values” has not the same font size as other words. The same problem appears to be below equation 2.

Response: 

Comment has been implemented accordingly.

Comment:

The table 4 of page 12： The word “Increase” appears in two lines many times.

Response: 

Comment has been implemented accordingly.

Comment: 

Line 10 of page 13: "there still exist a weak correlation" is error in grammar. And yhy line numbers begin this page and not in page 1-12?

Response: 

Comment has been implemented accordingly.

Comment: 

Line 38-39 of page 14: " Areas with historically less rainfall is projected to ......" is error in grammar.

Response: 

Sentence has been rephrased for clarity.

Comment: 

Line 86-87 of page 16: "It is anticipated that future temperature will increase, and rainfall decrease under the business-as- usual RCP8.5 scenario over the Olifants River Catchment." The description as “future temperature will increase in RCP8.5” does not make any sense in science

Response: 

Sentence has been rephrased for clarity.

Comment:

Line 87-88 of page 16: "The findings of this study as related to temperature agrees with the findings of " is wrong in grammar.

Response: 

Sentence has been rephrased for clarity.

Comment: 

Line 102-103 of page 16: "As temperatures rise and rainfall decreases, crop yield will be affected" is wrong in grammar.

Response: 

Sentence has been rephrased for clarity.

Comment: 

Line 118 of page 17: "An ensemble of eight regional climate model simulations of the CORDEX Africa forced with CMIP5 models " What are the eight regional climate model?

Response: 

Sentence has been rephrased for clarity.

Comment: 

The reference in page 19 should be list in the same format. The discrepancy is presented like "(2019)" of citation 2 and “2006”in 3. Please check all reference

Response: 

References has been checked for uniformity.

Thank you again for the positive feedback and thank you for your consideration. We look forward to receiving a good review of this paper.

Reviewer #3:

Comment: 

Require major revision , please see comments in attachment. This is a potentially very useful paper, but the current version and some work with the discussions and additional inputs in the methods and objectives.

Response: 

Thank you for the positive feedback. We appreciate the time and effort that the Reviewer has invested for understanding of our work and for the valuable comments for improvement of our manuscript. We appreciate the suggestions which help to improve the readability of our manuscript.

Comment: 

The methods and results are interesting, but the presentation of the manuscript is poor, particularly at the discussion level. Based on the discussion section, I think the manuscript is now more written as an internal report, and not a qualitative research paper. This study lacks a discussion section that in quantitative explaining the observed results. 

Response: 

The discussion section of the manuscript has been revised.

Comment: 

The conclusion actually is longer than the discussion not unless I am missing out on the whole context. This is quite a shame because the writing is really well done with logical flow with this information surely very useful in the advancement of the climate prediction agenda based on downscaled models.

Response: 

Comment is well noted. The conclusion section has been improved 

Comment: 

There is no clear objective and specific objectives, research question or hypothesis formulated. Instead, the authors need to come up with a clear research question, rethink the statistics they use on the data based on that question, elaborate in the methods section, and then write a focused discussion.

Response: 

The manuscript has been revised and generally improved.

Thank you again for the positive feedback and thank you for your consideration. We look forward to receiving a good review of this paper. 

Reviewer #4A: 

Comment: 

The authors attempt to predict the future trends and projections of surface temperature and rainfall over the Olifants River catchment using downscaling. 

Response: 

The authors thank you for the quality review of their work. We have tried to implements all comments and we are sure of returning to you a much improved manuscript.

Comment: 

The manuscript lacks coherent structure and suffers from methodological gaps and unfounded conclusions. 

Response: 

Comments are noted and the methodology and conclusions have been improved 

Comment: 

The paper has a medium quality in terms of the language, with various syntax mistakes (extra checks are required). 

Response: 

Manuscript has been improved for clarity

Comment: 

The introduction contains a lot of information, but fails to provide the background intelligible and outline the overall scope of the study adequately.

Response: 

Comments are noted and have been implemented

Comment: 

When it comes to the description of the methodology, many confusions arose in the ‘data set’ portion as it was found difficult to identify which data set was used for the present study. 

Response: 

The methodology section has been improved as suggested

Comment: 

Please add further explanation for the experimental design (in the methods section). 

Response: 

The methodology section has been improved as suggested

Comment: 

In addition, when the readers reach the results and conclusions section, poor presentation (in terms of image resolution and position) of Figures made it difficult to identify the findings. 

Response: 

Improved figures have been privided

Comment: 

Summarily, this manuscript seems to be a simple effort by the authors to present future climates projected by CMIP5 models. 

Response: 

The comments are well noted authors have improved the manuscript

Comment: 

Based on the current state of the manuscript, I suggest that the research should only be accepted for publication if the authors complete the major revisions suggested in the uploaded manuscript.

Response: 

Thank you again for the positive feedback and thank you for your consideration. We look forward to receiving a good review of this paper. If you need further information, please feel free to contact me.

Comment: 

Since, line number is not given properly; I did my corrections in the submitted PDF. Kindly go through it and do the necessary corrections.

Response: 

Response:

The manuscript has generally been updated. Thank you again for the positive feedback and thank you for your consideration. We look forward to receiving a good review of this paper.

#Reviewer 4B

Title: Trends and projections of surface temperature and rainfall over the Olifants River catchment

General comments

This is a potentially very useful paper, but the current version and some work with the discussions and additional inputs in the methods and objectives.

The methods and results are interesting, but the presentation of the manuscript is poor, particularly at the discussion level. Based on the discussion section, I think the manuscript is now more written as an internal report, and not a qualitative research paper. This study lacks a discussion section that in quantitative explaining the observed results. The conclusion actually is longer than the discussion not unless I am missing out on the whole context. This is quite a shame because the writing is really well done with logical flow with this information surely very useful in the advancement of the climate prediction agenda based on downscaled models.

Comment:

There is no clear objective and specific objectives, research question or hypothesis formulated. Instead, the authors need to come up with a clear research question, rethink the statistics they use on the data based on that question, elaborate in the methods section, and then write a focused discussion. 

Response:

The comments are well noted and have been implemented in the revised version of the manuscript

COMMENTS ON MANUSCRIPT; 

Given the title, (Title Trends and projections of surface temperature and rainfall over the Olifants River catchment) and also given the approaches used in the downscaling of data, I think the title is inefficient. I suggest inclusion of the models, i.e., like the following below, 

Trends and projections of surface temperature and rainfall over the Olifants River catchment in (Country ?) CMIP5 model ensemble?

Response:

The comments are well noted and the title has been changed accordingly

ABSTRACT

•The abstract is well written covering where and how the research was carried out. The abstract also shows the main findings from the study. 

Introduction

• This is well written showing the objectives of the study. et al

Objective

The authors have stated the objective as follows,

This study focusses on investigating the trends in historical maximum, minimum temperatures and rainfall, the estimation of plausible future climate changes, as well as the magnitude of future occurrences in the variables to inform adaptation initiatives over the Olifants River Catchment. I think this objective appears very general and lacks specifics and maybe that why the discussion section has been fluffed. I suggest the following,

Include the period in context when capturing the general and specific objectives,

• Also include the ensemble in question in the objectives e.g., dynamically downscaled GCM ensemble using the CMIP5 dataset

• Also include research questions, /hypothesis to be tested.

• the RCP4.5 and RCP8.5 scenarios should also be captured in your objectives. In the absence readers will be lost on what scenarios re been evaluated.

Response:

The comments are well noted and have been implemented in the revised version of the manuscript

Table 3,4: Descriptive statistics and trend analysis of maximum, minimum temperature (°C) and rainfall (mm) is better illustrated in a graph.

Material and methods

Kindly note that the study area should contain the following information, 

Geographical location and description (inform the reader whether area is gazetted, protected or the form of management), climatic information, vegetation types, drainage information, demographic features, land use and resources in an abridged form.

Response:

The comments are well noted and the study area section has been revised

Th authors should justify why the picked on the ensemble of eight individual Global Climate Models (GCMs) that participated in the Fifth Phase of Coupled Model Inter-comparison Project (CMIP5). Were there more that participated? Why the eight why not 5? Proper justification required.

Response:

The comments are well noted and general update and justification has been done

I have a problem with picking historical data for only two stations out of the 4 which basically means 50 percent data availability. This surely must compromise the data.

Response:

The comments are well noted and justification and general rephrasing of the section has been done to provide better clarity

I would like to see the the standardised precipitation index (SPI) classification for extreme weather events (and probability of occurrence) which then can be illustrated through a graph.

Response:

The comments are well noted and SPI 2,6 and 12-months have been included in the revised version

Future projections of rainfall, maximum and minimum temperature are presented for the three 30-year periods extending from 2036 – 2065 (near future) and 2066 – 2095 (far future) under the RCP 8.5 scenario. Projected changes are expressed relative to the historical 30-year period of 1976 to 2005. . Based on the above assertions its baffling that the authors limited themselves to baseline scenarios and worst-case scenario (RCP8.5) yet we are living in the intermediate scenario (RCP4.5). Why did the authors omit the RCP 4.5 scenario? We also have RCP 6.5.

Response:

The comments are well noted and RCP4.5 has been added in the revised version

Results 

Results have been well enumerated, however with the absence of aspects captured in the objectives and methods section. It is still inadequate. Most of the figures are not clear and are not easy to comprehend. A simple graphical presentation of rainfall patterns, historical and projected gives a very simple but clearer picture of the peaks, extreme events and also the linear line give a good impression of the trends. This should be adopted for the seasonal, decadal, annual analysis as well.

Response:

The comments are well noted and all figures have been improved and well illustrated

Discussion

Based on the discussion section, I think the manuscript is now more written as an internal report, and not a qualitative research paper. This study lacks a discussion section that in quantitative explaining the observed results. The conclusion actually is longer than the discussion not unless I am missing out on the whole context. This is quite a shame because the writing is really well done with logical flow with this information surely very useful in the advancement of the climate prediction agenda based on downscaled models. As you enhance your discussion, critically assess the output for the model’s vis a vis extreme rainfall events both for the annual, decadal and seasonal. Also establish whether the models are biased for the short rains over the long rains or vice versa. Compare your results to the east African models which have been christened the “East African climate paradox” as alluded by Rowel et al. 2015, where the observed rainfall trends exhibited a drying anomaly compared to the scenario simulation which projects a wetting anomaly.

In the discussions, I would like to see the following, discussion points on

i Historical rainfall and temperature trends

ii Historical seasonal trends

iii Extreme rainfall events historical in nature

iv Projected rainfall (including seasonality) and temperature trends based on the RCP 4.5 and 8.5

v Projected Extreme rainfall events (forecast on droughts, floods, storms etc)

vi Efficacy of the downscaled models I projecting rainfall but importantly mimicking historical rainfall which is the basis for the projection.

Response:

The comments are well noted and the discussions section of the manuscript has been revised

Conclusion and Recommendation

 I have reserved my comments for conclusion and recommendation because according to this paper, the discussion section is missing. it is impossible to make conclusions.

Response:

The manuscript has generally been updated. Thank you again for the positive feedback and thank you for your consideration. We look forward to receiving a good review of this paper.

---

## [Decision Letter · Decision Letter 1]

4 Apr 2022

PONE-D-21-31259R1Observed Trends and Projections of Temperature and Precipitation in the Olifants River Catchment in South AfricaPLOS ONE

Dear Dr. Adeola,

Thank you for submitting your manuscript to PLOS ONE. After careful consideration, we feel that it has merit but does not fully meet PLOS ONE’s publication criteria as it currently stands. Therefore, we invite you to submit a revised version of the manuscript that addresses the points raised during the review process.

ACADEMIC EDITOR:  The authors have addressed most of the reviewers' comments. However, the article still needs some improvement on the discussion section to make it up to the standard of publication. Please look at the reviewers' comments carefully and made the changes accordingly. Thank you.  

We look forward to receiving your revised manuscript.

Kind regards,

Mou Leong Tan

Academic Editor

PLOS ONE

Journal Requirements:

Additional Editor Comments (if provided): 

The authors have addressed most of the reviewers' comments. However, the article still needs some improvement on the discussion section to make it up to the standard of publication. Thank you.

Reviewers' comments:

Reviewer's Responses to Questions

**Comments to the Author**

1. If the authors have adequately addressed your comments raised in a previous round of review and you feel that this manuscript is now acceptable for publication, you may indicate that here to bypass the “Comments to the Author” section, enter your conflict of interest statement in the “Confidential to Editor” section, and submit your "Accept" recommendation.

Reviewer #1: All comments have been addressed

Reviewer #2: All comments have been addressed

Reviewer #3: (No Response)

Reviewer #4: All comments have been addressed

2. Is the manuscript technically sound, and do the data support the conclusions?

Reviewer #1: Yes

Reviewer #2: Yes

Reviewer #3: Yes

Reviewer #4: Yes

3. Has the statistical analysis been performed appropriately and rigorously? 

Reviewer #1: Yes

Reviewer #2: Yes

Reviewer #3: Yes

Reviewer #4: Yes

4. Have the authors made all data underlying the findings in their manuscript fully available?

Reviewer #1: No

Reviewer #2: Yes

Reviewer #3: Yes

Reviewer #4: Yes

5. Is the manuscript presented in an intelligible fashion and written in standard English?

Reviewer #1: No

Reviewer #2: Yes

Reviewer #3: Yes

Reviewer #4: Yes

6. Review Comments to the Author

Reviewer #1: The authors have effectively considered most of my comments. I appreciate the analysis of a more moderate emission pathway (RCP4.5) and the discussion on the significance of trends. Before recommending publication, I would like to propose a very brief comparison with CMIP6 results (e.g., from Almazroui et al. 2020, or the IPCC AR6 atlas of regional changes(https://interactive-atlas.ipcc.ch/)). This could be a part of the discussion. Some additional effort is also needed towards improving the readability of the manuscript.

References

Almazroui, M., Saeed, F., Saeed, S., Nazrul Islam, M., Ismail, M., Klutse, N.A.B., Siddiqui, M.H., 2020. Projected Change in Temperature and Precipitation Over Africa from CMIP6. Earth Syst. Environ. 4, 455–475. https://doi.org/10.1007/s41748-020-00161-x

Reviewer #2: The authors have done the revisions as suggested in the previous round. So there are no more suggestions now.

Reviewer #3: Please look at my comments on the discussions, You have amazing data and have done some excellent analysis but the discussion botch up the whole paper. You have many discussion points as per my comments on that section ink man last submission. I encourage that you follow those item by time. I still feel that the discussion section is really exposed and won't do justice to the paper if accepted for publication

Reviewer #4: The manuscript has been improved a lot as per the suggestion of the reviewers. It is now ready for acceptance (if other reviewers have similar views).

7. PLOS authors have the option to publish the peer review history of their article (what does this mean?). If published, this will include your full peer review and any attached files.

Reviewer #1: No

Reviewer #2: No

Reviewer #3: No

Reviewer #4: No

---

## [Author Response · Author response to Decision Letter 1]

31 May 2022

Reviewer #1: The authors have effectively considered most of my comments. I appreciate the analysis of a more moderate emission pathway (RCP4.5) and the discussion on the significance of trends. Before recommending publication, I would like to propose a very brief comparison with CMIP6 results (e.g., from Almazroui et al. 2020, or the IPCC AR6 atlas of regional changes(https://interactive-atlas.ipcc.ch/)). This could be a part of the discussion. Some additional effort is also needed towards improving the readability of the manuscript.

References

Almazroui, M., Saeed, F., Saeed, S., Nazrul Islam, M., Ismail, M., Klutse, N.A.B., Siddiqui, M.H., 2020. Projected Change in Temperature and Precipitation Over Africa from CMIP6. Earth Syst. Environ. 4, 455–475. https://doi.org/10.1007/s41748-020-00161-x

Response: 

We thank the Reviewer to this valuable comment. We have implemented the comment and we are glad to return a much improved discussion section in the revised manuscript.

Reviewer #2: The authors have done the revisions as suggested in the previous round. So there are no more suggestions now.

Response: 

We thank the Reviewer for the valuable comments that helped improved the manuscript and we are glad that they are happy with the revised version.

Reviewer #3: Please look at my comments on the discussions, You have amazing data and have done some excellent analysis but the discussion botch up the whole paper. You have many discussion points as per my comments on that section ink man last submission. I encourage that you follow those item by time. I still feel that the discussion section is really exposed and won't do justice to the paper if accepted for publication

Response: 

We thank the Reviewer to this valuable comment. We have implemented the comment and we are glad to return a much improved discussion section in the revised manuscript.

Reviewer #4: The manuscript has been improved a lot as per the suggestion of the reviewers. It is now ready for acceptance (if other reviewers have similar views).

Response: 

We thank the Reviewer to this valuable comment. We have implemented the comment and we are glad to return a much improved discussion section in the revised manuscript.

---

## [Editor Report · Decision Letter 2]

12 Jul 2022

Observed Trends and Projections of Temperature and Precipitation in the Olifants River Catchment in South Africa

PONE-D-21-31259R2

Dear Dr. Adeola,

We’re pleased to inform you that your manuscript has been judged scientifically suitable for publication and will be formally accepted for publication once it meets all outstanding technical requirements.

Kind regards,

Mou Leong Tan

Academic Editor

PLOS ONE

Additional Editor Comments (optional):

The authors have improved their discussion based on the suggestion. The manuscript is now suitable for publication in Plos One. Thank you.
---

## [Editor Report · Acceptance letter]

1 Aug 2022

PONE-D-21-31259R2 

Observed Trends and Projections of Temperature and Precipitation in the Olifants River Catchment in South Africa 

Dear Dr. Adeola:

I'm pleased to inform you that your manuscript has been deemed suitable for publication in PLOS ONE. Congratulations! Your manuscript is now with our production department. 

Kind regards, 

on behalf of

Dr. Mou Leong Tan 

Academic Editor

PLOS ONE